# Telehealth Intervention to Improve Uptake of Evidence-Based Medications among Patients with Type 2 Diabetes and Heart Failure or Cardiovascular Disease

**DOI:** 10.3390/ijerph20043613

**Published:** 2023-02-17

**Authors:** Armando Silva-Almodóvar, Milap C. Nahata

**Affiliations:** 1Institute of Therapeutic Innovations and Outcomes (ITIO), College of Pharmacy, The Ohio State University, Columbus, OH 43210, USA; 2College of Medicine, The Ohio State University, Columbus, OH 43210, USA

**Keywords:** diabetes mellitus, heart failure, cardiovascular disease, medication therapy management, targeted medication review, Sodium glucose cotransporter-2 inhibitors, glucagon-like peptide 1 receptor agonists

## Abstract

Introduction: Sodium glucose cotransporter-2 (SGLT-2) inhibitors and glucagon-like peptide 1 receptor (GLP-1) agonists are recommended for patients with type two diabetes (T2D) and atherosclerotic cardiovascular disease (ASCVD) or heart failure (HF) to reduce cardiovascular-related mortality. The objective of this study was to evaluate a telehealth targeted medication review (TMR) program to identify patients for uptake of these evidence-based medications. Methods: This was an observational descriptive study of a TMR program for Medicare-enrolled, Medication Therapy Management-eligible patients in one insurance plan. Prescription claims and patient interviews identified individuals who would benefit from SGLT-2 inhibitors or GLP-1 agonists. Facsimiles were sent to providers of patients with educational information about the targeted medications. Descriptive statistics described characteristics and proportion of patients prescribed targeted medications after 120 days. Bivariate statistical tests evaluated associations between age, sex, number of medications, number of providers, and poverty level with adoption of targeted medications. Results: A total of 1106 of 1127 had a facsimile sent to their provider after a conversation with the patient. Among patients with a provider facsimile, 69 (6%) patients filled a prescription for a targeted medication after 120 days. There was a significant difference in age between individuals who started a targeted medication (67 ± 10 years) compared with patients who did not (71 ± 10 years) (*p* = 0.001). Conclusions: A TMR efficiently identified patients with T2D and ASCVD or HF who would benefit from evidence-based medications. Although younger patients were more likely to receive these medications, the overall uptake of these medications within four months of the intervention was lower than expected.

## 1. Introduction

It is estimated that as many as 50% of older adults with diabetes are at very high risk for cardiovascular-related mortality [1]. Sodium glucose cotransporter 2 inhibitors (SGLT-2) and glucagon-like peptide 1 receptor (GLP-1) agonists have proven benefits in the treatment of atherosclerotic cardiovascular disease (ASCVD) and are indicated for patients co-diagnosed with type 2 diabetes (T2D) and ASCVD, at high risk for ASCVD, or with heart failure (HF) irrespective of surrogate markers of diabetes control [2]. However, despite the importance of these medications in reducing long-term healthcare utilization among qualifying patients, uptake of these evidence-based medicines has been low. Only 0–20% of patients enrolled in various insurance programs (commercial, Medicare, Medicaid, or Veteran’s Affairs) were receiving these medications in the United States (US) [3,4,5]. 

Importantly, the prescribing of these medications has increased among cardiologists over time [6]. However, one survey conducted among general practitioners found that underprescribing of these medications could be related to an under-recognition of the benefits of these medications, lack of access to endocrinologists for consultation, and concerns with the medicines’ associated adverse events [7]. To facilitate prescribing, some academic centers have incorporated educational interventions to increase awareness of the benefits of these medications among prescribers [8]. Furthermore, it has been suggested that clinical decision algorithms within healthcare settings may increase prescribing of these medications for at-risk patients [9]. However, to date, there are no published interventions that can enhance identification of patients across healthcare systems who would benefit from these medications.

Medication Therapy Management (MTM) is one means of enhancing uptake of evidence-based medications. Through comprehensive medication reviews (CMRs) and targeted medication review (TMRs), MTM programs identify patients who may benefit from changes to their therapeutic regimens [10]. These programs incorporate structured patient interviews and electronic monitoring of healthcare data to identify opportunities for improving medication regimens. This study sought to describe and evaluate a low-cost telehealth TMR program that electronically identified patients with T2D and ASCVD or HF utilizing prescription claims to increase uptake of evidence-based medications.

## 2. Methods

### 2.1. Study Design

This was a retrospective observational descriptive study of a TMR program that was utilized for patients in one Medicare plan. For this TMR, patients were first identified with prescription claims that indicated a possible diagnosis of T2D and ASCVD or HF. Patients were subsequently contacted telephonically by pharmacists to verify the diagnoses utilizing a standardized script and to ascertain the absence of contraindications for the prescribing of SGLT-2 inhibitors or GLP-1 agonists. Once they were confirmed as being eligible for the targeted medications, a facsimile providing important education on the benefits of these medications for this population along with a recommendation to initiate this medication was sent to their prescribers. Prescription claims were evaluated 120 days after the intervention to determine if the recommendation was successful as documented by the addition of an SGLT-2 inhibitor or a GLP-1 agonist to the patient’s medication regimen. 

### 2.2. Study Population and Data Collection

This study included patients who were contacted between 1 January 2021, and 20 November 2021. Patients were included if they were Medicare-enrolled, MTM-eligible, received the TMR service, were deemed eligible for an SGLT-1 inhibitor or a GLP-1 agonist, if a facsimile was sent to their provider, and if 120 days had passed since the delivery of the intervention at the time of data collection. Patients who participated in Medicare-enrolled plans were older than 65 years of age. Certain patients under 65 years of age with Medicare insurance qualified due to being disabled or diagnosed with end-stage renal disease or amyotrophic lateral sclerosis [11].

### 2.3. Statistical Analysis

Data were coded and organized using Microsoft Excel (2016 MSO, Redmond, WA, USA) and IBM SPSS software (v28.0, IBM Corp, Armonk, NY). Counts and percentages and means and standard deviations were used to describe variables as appropriate. Variables collected included age (continuous), sex (male, female), ZIP code (continuous), number of prescribers (count), number of medications (count), and if an individual experienced a medication change after 120 days (yes, no). The ZIP code was cross-referenced with data provided by the Census Bureau to estimate the percent of individuals that were below the federal poverty line in a specific ZIP code [12]. It was also used to determine what region within the US a patient lived [13]. An exploratory analysis was conducted via Chi-square and Fischer Exact tests to evaluate if there were any detectable associations between patient characteristics and adoption of the targeted medications. A post hoc t-test was performed to evaluate if there was a significant difference in age between individuals who were prescribed the evidence-based medications and those who were not. A p-value of 0.05 determined statistical significance.

## 3. Results

In this study, 1442 patients were initially identified as potentially benefitting from an SLGT-2 inhibitor or GLP-1 agonist. At the time of the analysis, 315 patients were excluded due to insufficient time having passed to assess the outcome of the alert. After review of the clinical profile and an interview with the patient, 21 patients did not have a facsimile sent to their provider. For this analysis, 1106 patients had a facsimile sent to their provider and were included in this study. Patients were 71 ± 10 years of age on average, predominantly female (602, 54%), filled 13 ± 4 unique medications to manage chronic conditions, and utilized 3 ± 2 prescribers. In this study, 638 (62%) patients lived in the South, 237 (23%) lived in the West, 100 (10%) lived in the Midwest, and 61 (6%) lived in the Northeast regions of the US.

After 120 days, 69 (6%) patients filled a prescription for an SGLT-2 inhibitor or GLP-1 agonist utilizing their insurance benefit. Among all assessed variables, the only variable associated with adoption of the targeted medications was age (*p* = 0.006). There was a significant difference in the age between individuals who started a targeted medication (67 ± 10 years) compared with patients who did not (71 ± 10 years) (*p* = 0.001). No other assessed variable was individually associated with initiating the targeted medications. Complete results are described in Table 1.

## 4. Discussion

As many as 50% of patients with diabetes may be at high risk for cardiovascular related mortality [1]. Previous research has shown that most patients with diabetes who would benefit from evidence-based medications to reduce the risk of cardiovascular related mortality were not prescribed an SGLT-2 inhibitor or a GLP-1 agonist [3,4,5]. To increase prescribing of these medications among this vulnerable cohort it may be important to facilitate the identification of which patients benefit from the prescribing of these medications. Interventions to increase uptake of these medications are needed to reduce healthcare utilization in this at-risk cohort. This study demonstrated a TMR telehealth facsimile intervention was capable of efficiently identifying patients with DM and CVD or HF who would benefit from evidence-based medications expected to lower long-term healthcare utilization. However, the uptake of these medications within four months of the intervention was lower than expected. This study also found younger patients were significantly more likely to receive these targeted medications than the older population within the observation period. It has been suggested that older patients may be more susceptible to clinical therapeutic inertia and may be less likely to understand the benefits of these medications in improving glucose control to reduce potential healthcare utilization [14,15].

While this study explored living in areas of increasing poverty level as a factor in affecting uptake of costly medications, this variable was not associated with receipt of receiving the targeted medications. This may have occurred for several reasons. In the US, Medicare patients are enrolled in health plans where drugs will cost a fixed price or will cost a percentage of the overall cost of a drug to the patients [16]. Patients can also purchase additional insurance coverage or those who meet specific income and poverty requirements may benefit from state and federal programs to lower the cost of their medications to them [17]. These factors may be significantly more important when considering the addition of an expensive brand medication to a patient’s therapeutic regimen in comparison to living in ZIP codes with increasing levels of poverty. 

In addition to patient-related factors, insurance plans can also unwittingly limit the uptake of evidence-based medications. For example, insurance plans may utilize prior authorizations, formulary restrictions, high coinsurance, and step therapy requirements to approve the use of certain high-cost branded medications by patients and thus reduce the costs to the insurance company. These regulatory barriers can delay the prescribing and usage of brand medications among targeted patients [18]. This is problematic as patients with T2D are already at an increased risk of high out-of-pocket healthcare costs resulting in cost-related nonadherence when patients may not be able to afford to pay for additional costly medications [19]. Future research should seek patient and provider feedback to determine reasons for denying the addition of an SGLT-2 inhibitor or GLP-1 agonist to a therapeutic regimen. Furthermore, evaluation of this intervention should determine the costs avoided by the insurance plan over time given that increased uptake of these medications is expected to reduce long term healthcare utilization and costs. 

### Limitations

There are important limitations to consider for this study. Investigators were unable to assess prescriber rationale for not adding the evidence-based targeted medication to a patient’s medication regimen. Despite communication with patients about the need for these medications, investigators were unable to confirm if all providers received and understood the content of the facsimiles that were sent regarding the reason for the intervention. The four-month follow-up period may have been too short to fully observe the implementation of the intervention. Authors were also unable to evaluate the influence of disease burden and cost of medications on uptake of the intervention. 

## 5. Conclusions

Telehealth TMR interventions efficiently identified patients with T2D in need of prescribing of evidence-based medicines. This study successfully identified patients from an evaluation of prescription claims and patient interviews for their potential need for an SGLT-2 inhibitor or GLP-1 agonist. Patients who received the targeted medications were significantly younger than patients who did not during the observation period. The uptake of evidence-based medications in this study, however, was markedly lower than expected. Future investigations should seek healthcare provider and patient feedback to evaluate reasons for declining to add SGLT-2 inhibitors or GLP-1 agonists to a patient’s therapeutic regimen.

## Figures and Tables

**Table 1 ijerph-20-03613-t001:** Characteristics of patients included in this study.

Characteristics	All Patients (N: 1106)	Patients with a Medication Change (N:69)	Patients without Medication Change (N:1037)	*p*-Value
	N (%)	N (%)	N (%)	
Sex				
Female	602 (54)	30 (44)	572 (55)	0.06
Male	504 (46)	39 (57)	465 (45)	
Age, years				
<65	288 (26)	29 (42)	259 (25)	0.006
65–74	401 (36)	23 (33)	378 (37)	
75–84	330 (30)	16 (23)	314 (30)	
>84	87 (8)	1 (1)	86 (8)	
Percent of individuals living below poverty level in an individual’s ZIP code				
0–9.99	262 (24)	17 (25)	245 (24)	0.60
10–19.99	495 (45)	35 (51)	460 (44)	
20–29.99	282 (26)	13 (19)	269 (26)	
≥30	65 (6)	4 (6)	61 (6)	
Number of Unique Medications				
5–8	113 (10)	7 (10)	106 (10)	0.98
9–12	475 (43)	28 (41)	447 (43)	
13–16	341 (31)	22 (32)	319 (31)	
≥17	177 (16)	12 (17)	165 (16)	
Number of Unique Prescribers				
1	210 (19)	10 (15)	200 (19)	0.86
2	243 (22)	17 (25)	226 (22)	
3	233 (21)	14 (20)	219 (21)	
4	162 (15)	10 (15)	152 (15)	
≥5	258 (23)	18 (26)	240 (23)	

## Data Availability

Data are available from the authors upon reasonable request and with permission of MedwiseRx.

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
