# Peer review of "Telehealth Intervention to Improve Uptake of Evidence-Based Medications among Patients with Type 2 Diabetes and Heart Failure or Cardiovascular Disease"

_ijerph, 2023, doi:10.3390/ijerph20043613_

Round 1

Reviewer 1 Report

1) Authors must follow the instruction for user especially on references citations.

2) Why the average age of patients used for the current study was 71+/- 10 , You must screen more age starting from 30 to 80 years old age.

3) From my point of view this study need more time to give more implementation for the intervention.

Author Response

1) Authors must follow the instruction for user especially on references citations.

Authors have made edits to the citations to fit the recommended guidance.

2) Why the average age of patients used for the current study was 71+/-10, You must screen more age starting from 30 to 80 years old age.

This study is describing a service that was provided to patients that were enrolled in a Medicare insurance plan. Authors have included the following under the section Study population and data collection in the Methods:

“Patients who participated in Medicare enrolled plans were older than 65 years of age. Patients under 65 years of age on Medicare qualified due to being disabled or diagnosed with end-stage renal disease or amyotrophic lateral sclerosis.”

It is not possible to explicitly recruit patients between the ages of 30-80 given the patient population we served. Our study was explicitly designed to evaluate Medicare patients in the age group reported in this paper.

3) From my point of view this study need more time to give more implementation for the intervention.

As mentioned in the Limitations, the four-month follow-up period may potentially contribute to the low uptake of the intervention. However, four-month follow-up is currently used by most insurance providers, the organizations that request these clinical services. We are evaluating extending the evaluation period for follow-up studies.

Reviewer 2 Report

It is an interesting study that provides a new perspective to understand how telehealth services can electronically identify patients who would benefit from using evidence-based medications.

However, some revisions are needed:

a. In the manuscript, the authors do not specify if there is an informed consent statement of the patient participating in the study.

b. In terms of statistical analysis, I believe that few variables were evaluated to determine whether the recommendation of SGLT-2 inhibitors or GLP-1 agonists was successful.

I don't know if there are additional data regarding the analysis of other variables.

If they exist, I suggest the authors to present them to enrich the results of the study and bring additional evidence in support of the benefits of telehealth services in the selection of potential patients benefiting from evidence-based medications.

Author Response

It is an interesting study that provides a new perspective to understand how telehealth services can electronically identify patients who would benefit from using evidence-based medications.

Authors appreciate the interest.

However, some revisions are needed:

In the manuscript, the authors do not specify if there is an informed consent statement of the patient participating in the study.

Authors have sought the necessary approvals for conducting research by submitting the necessary documentation to the Institutional Review Board as noted at the end of the text on page 5. Informed consent requirements were waived. This project was a retrospective review of information captured within an electronic pharmacy record and as such did not require an informed consent.

b. In terms of statistical analysis, I believe that few variables were evaluated to determine whether the recommendation of SGLT-2 inhibitors or GLP-1 agonists was successful.

I don't know if there are additional data regarding the analysis of other variables.

If they exist, I suggest the authors to present them to enrich the results of the study and bring additional evidence in support of the benefits of telehealth services in the selection of potential patients benefiting from evidence-based medications.

Authors have used all variables that they had access to. Authors have included in the Limitations we were unable to evaluate the effects of disease burden and cost of medications on uptake of the intervention.

Round 2

Reviewer 2 Report

Since the authors used all variables to which they had access (according to the response received) and this project is a retrospective analysis of information from the pharmacy's electronic database, a situation that does not require obtaining the patient's informed consent, I agree with the acceptance of the work in the present form.

At the same time, I suggest to the authors that in the future, on the occasion of other publications, to evaluate more variables to enrich the results of the study and increase the scientific value of the work.